# CD32 expression is associated to T-cell activation and is not a marker of the HIV-1 reservoir

Roger Badia[1], Ester Ballana[1], Marc Castellví[1], Edurne García-Vidal[1], Maria Pujantell[1], Bonaventura Clotet[1], Julia G. Prado[1], Jordi Puig[1], Miguel A. Martínez [1], Eva Riveira-Muñoz[1] & José A. Esté [1]

CD32 has been shown to be preferentially expressed in latently HIV-1-infected cells in an in vitro model of quiescent CD4 T cells. Here we show that stimulation of CD4+ T cells with IL-2, IL-7, PHA, and anti-CD3/CD28 antibodies induces T-cell proliferation, co-expression of CD32 and the activation of the markers HLA-DR and CD69. HIV-1 infection increases CD32 expression. 79.2% of the CD32+/CD4+ T cells from HIV+ individuals under antiretroviral treatment were HLA-DR+. Resting CD4+ T cells infected in vitro generally results in higher integration of provirus. We observe no difference in provirus integration or replication-competent inducible latent HIV-1 in CD32+ or CD32− CD4+ T cells from HIV+ individuals. Our results demonstrate that CD32 expression is a marker of CD4+ T cell activation in HIV+ individuals and raises questions regarding the immune resting status of CD32+ cells harboring HIV-1 proviruses.

[1] AIDS Research Institute-IrsiCaixa and Health Research Institute Germans Trias i Pujol (IGTP), Hospital Germans Trias i Pujol, Universitat Autònoma de Barcelona, 08916 Badalona, Spain. These authors contributed equally: Roger Badia, Ester Ballana. Correspondence and requests for materials should be addressed to E.R.-Mño. (email: eriveira@irsicaixa.es) or to J.A.Eé. (email: jaeste@irsicaixa.es)

The use of antiretroviral therapy (ART) has significantly transformed HIV-1 infection from a terminal illness to a chronic manageable disease[1]. Despite intensive investigation, no strategy to date has resulted in sustained control of HIV in the absence of ART.

HIV-1 infects activated CD4+ T cells and results in active virus replication or immediate silent integration[2]. Latency is established within a narrow time window after activation[3] or during the transition of these HIV-infected and activated cells to resting memory CD4+ T cells[4].

Eisele and Silicano define the HIV-1 reservoir as an infected cell population that allows the persistence of replication-competent HIV-1 in patients on optimal treatment regimens on a timescale of years. To date, the latent reservoir in resting CD4+ T cells is the only reservoir shown to fit this definition[5]. The International AIDS Society Scientific Working Group on HIV Cure has suggested that the best characterized and only proven cellular reservoir of HIV during long-term HIV treatment are memory CD4+ T cells that lack activation markers[6]. Indeed, latently infected resting memory CD4+ T cells form the largest HIV-1 reservoir and represent the subset with the greatest clinical importance because of their long lifespan[5]. The quest for long-term control of HIV-1 in the absence of ART has led to numerous therapeutic approaches aimed at increasing host-mediated control of HIV-1 or clearance of latent virus reservoirs[7–9] while maintaining the beneficial effects of immune reconstitution.

Cells latently infected with HIV-1 are not thought to produce viral proteins and have long been considered indistinguishable from uninfected cells for all practical purposes[10]. Molecular signatures that allow for the identification of resting, latently infected cells would facilitate the study of HIV latency and accelerate the generation of new insights and therapeutic approaches[11]. Recently, Descours et al.[12] showed the overexpression of 103 differentially expressed genes, including 16 that encode transmembrane proteins, in apparently HIV+ resting cells in culture. The most highly expressed gene was FCGR2A, which encodes the Fc-gamma receptor FcγR-IIa (CD32a). The authors showed that CD32a+ cells from HIV-1+ participants were enriched in HIV DNA and inducible replication competent virus and concluded that CD32a is a cell surface marker of the CD4+ T cell HIV reservoir in HIV-infected virally suppressed participants.

FcγR represents a link between the humoral and cellular immune responses by triggering several functions, such as endocytosis and antibody-dependent cell-mediated cytotoxicity (ADCC)[13]. CD32a is a low-affinity Fc receptor with specificity for IgG antibodies and is commonly expressed on most myeloid cells, including monocytes, macrophages, and eosinophils[14,15] and is also expressed in natural killer (NK) cells and B-lymphocytes[16,17]. CD32 is highly regulated by agents such as phorbol 12-myristate 13-acetate (PMA) and cytokines, including interferon gamma (IFN-γ), dexamethasone and granulocyte-macrophage colony stimulatory factor (GM-CSF)[14]. The regulation of innate immune response recruitment is an important function of IgG-binding receptors such as CD32[18]. In particular, CD32 triggers phagocytosis and ADCC, which explains the constitutive expression in macrophages and NK cells[14,15]. Notably, CD32 was shown to be significantly downregulated on the surface of multiple innate immune cell subsets in both treated and untreated HIV-1 infections. This downregulation could result in irreversibly reduced ADCC activity in progressive infection, even in the absence of active viral replication[18,19].

The finding that CD32 expression is a marker of a CD4+ T cell HIV-1 reservoir would likely significantly impact the development of cure-focused HIV diagnostics and treatments[20]. This possibility deserves careful consideration. CD32 is possibly the most thoroughly studied FcγR[14]; therefore, it represents an excellent diagnostic tool.

Here we show that CD32 is not a marker of the HIV-1 reservoir. We explored CD32 expression following ex vivo HIV-1 infection in activated and resting cells or in peripheral blood mononuclear cells (PBMCs) from HIV+ individuals. CD32 expression is a marker of activation in a subset of CD4+ T cells both in uninfected controls and HIV+ individuals. We found that CD32+ cells are not enriched with integrated provirus DNA in the majority of HIV+ individuals receiving ART.

## Results

**CD32 is a marker of T-cell activation**. We evaluated CD32 expression in purified primary CD4+ T cells from HIV− control donors under different activating conditions, including PHA/IL-2, an anti-CD3/CD28 antibody and IL-2, IL-7 and IL-2, or IL-2 alone. CD4+ T cells were defined as CD3+/CD4+ cells or CD4+/CD8− cells depending on the activation stimulus used. The presence of conjugates between T cells and cells known to express high levels of CD32, such as CD19+ B cells or CD14+ monocytes[21], was excluded by gating on forward scatter (FSC) singlets and measuring the expression of CD19+ or CD14+ in the CD32+ cells and/or the CD4+ T-cell population (Supplementary Fig. 1a, b). Stimulation with IL-2, PHA/IL-2, anti-CD3/CD28/IL-2 and IL-7/IL-2-induced CD32+ expression as measured by flow cytometry (Fig. 1a, b, Supplementary Fig. 1b) and confirmed through the quantification of FCGR2A (CD32a) mRNA using qPCR in a subset of donor cells stimulated with IL-2 or PHA and IL-2 (Supplementary Fig. 2). CD32 expression was associated with cell proliferation as measured by intracellular Ki67 expression or T cell activation (Fig. 1a, c). Up to 80–90% of total CD32+ cells were HLA-DR+ when stimulated with PHA/IL-2, anti-CD3/CD28/IL-2, and IL-7/IL-2, and up to 75–80% were CD69+ when stimulated with PHA/IL-2 or IL-7/IL-2 (Fig. 1d). HLA-DR+ and CD69+ cells had upregulated CD32 expression compared with HLA-DR- or CD69-negative cells (Fig. 1e). As expected, CD32 was expressed in the majority of CD14+ monocytes (>90%) and CD19+ B cells (>90%) from uninfected donors (Supplementary Fig. 3).

CD4+ T cells from HIV+ individuals commonly express higher levels of T-cell activation markers, even after effective ART[22–25]. Because one of our main objectives was to evaluate CD32 expression and its relationship to HIV latency, we evaluated CD32 expression (according to the gating strategy described in Supplementary Fig. 4) in samples from HIV+ individuals under effective ART (viral load <50 HIV RNA copies/ml, Table 1). Additionally, isotype control labeling was set to a stringent criterion (≤0.1% positive cells) to avoid overestimating CD32 expression (Supplementary Fig. 5). The 0.1% marker sets the boundary of three standard deviations of a standard Gaussian distribution or a common standard in flow cytometry[26]. CD32 expression was significantly (Student's t-test, $p < 0.001$) higher in unstimulated CD4+ T cells from HIV+ individuals than uninfected, unstimulated donors (Fig. 2a), and it was highly associated with the activation marker HLA-DR but not with CD69 (Fig. 2b, c). This finding indicates a possible lack of functionality in T cells from HIV+ individuals[27,28]. A mean of 79.2% (70–94) of CD32+ cells were HLA-DR+ (Fig. 2d), indicating a strong correlation between CD32 expression and T-cell activation.

**CD32 expression following HIV-1 infection**. To further confirm the role of HIV-1 in CD32 expression, we evaluated cell surface marker expression following acute HIV-1 infection in cell culture. HIV-1 infection-induced CD32 expression in PHA/IL-2 activated

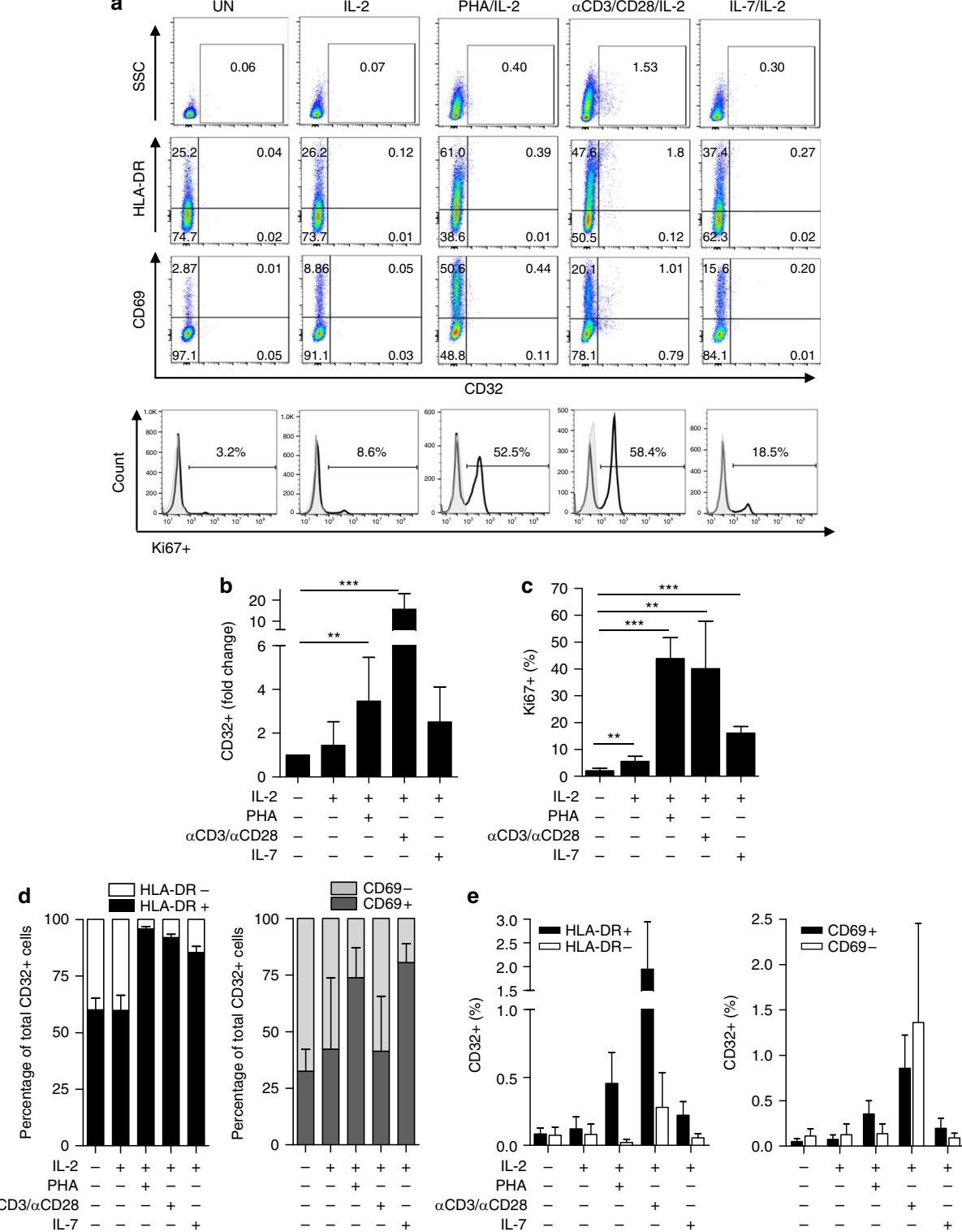

**Fig. 1** CD32 is a marker of T-cell activation. **a** Flow cytometry dot plots showing co-expression of CD32 and markers of cell activation and proliferation in unstimulated (UN) PBMCs or those stimulated with IL-2, PHA/IL-2, αCD3/αCD28/IL-2, and IL-7/IL-2. A representative donor is shown. **b** Fold change of CD32 expression in CD4+ T cells unstimulated or stimulated with different conditions from uninfected donors. The cells were cultured in the presence of different stimuli for 72 h, and protein levels of the cell surface marker CD32 were evaluated by flow cytometry. **c** Percentage of Ki67+ cells after activation with different stimuli as in (**a**). **d** Upregulation of CD32 correlates with the expression of activation markers HLA-DR and CD69 after activation with the different stimuli. Relative contribution of HLA-DR (left panel) or CD69 (right panel) cells over the total population of CD32-expressing cells. **e** Individual data of HLA-DR cells (left panel) or CD69 (right panel) cells in the CD32 compartment. The gating strategy used to identify CD32+ cells is shown in Supplementary Fig. 1. All panels represent the mean ± SD of at least five different donors. Student's *t*-test, *$p < 0.05$, **$p < 0.005$, ***$p < 0.0005$

## Table 1 Immunological and virological characteristics of HIV + individuals

| Patient number | CD4 (cells/μl) | CD8 (cells/μl) | VL (copies/ml) | Treatment[a] |
|---|---|---|---|---|
| P1 | 1746 | 1474 | <50 | ABC, DDI, NVP |
| P2 | 607 | 638 | <50 | ABC, 3TC, NVP, TDF |
| P3 | 877 | 1602 | <50 | D4T, IDV, 3TC, RTV |
| P4 | 796 | 1263 | <50 | ABC, LPV, NVP, RTV |
| P5 | 984 | 1650 | <50 | DDI, EFV, D4T, 3TC |
| P6 | 902 | 859 | <50 | EFV, 3TC, AZT |
| P7 | 987 | 740 | <50 | D4T, IDV, 3TC |
| P8 | 690 | 1074 | <50 | DDI, NVP, AZT |
| P9 | 935 | 1210 | <50 | IDV, 3TC, AZT |
| P10 | 382 | 870 | <50 | ABC, 3TC, RIL |
| P11 | 1340 | 1604 | <50 | DGT, MRV, RIL |
| P12 | 723 | NA | <50 | DTG, RIL |
| P13 | 234 | 620 | <50 | ELV, COBI, FTC |
| P14 | 279 | NA | <50 | DTG, RIL |
| P15 | 511 | 1235 | <50 | ABC, LPV, NVP, RTV |
| P16 | 719 | 464 | <50 | ELV, COBI, FTC, TAF |
| P17 | 340 | 1054 | <50 | EFV, TDF, FTC |
| P18 | 450 | 430 | <50 | DRV, COBI |
| P19 | 854 | 919 | <50 | ELV/C/F/TAF |
| P20 | 979 | 852 | <50 | RAL/TRU |
| P21 | 1240 | 897 | <50 | DTG/ABC/3TC |
| P22 | 897 | 481 | <50 | ELV/C/F/TAF |
| P23 | 943 | 849 | <50 | DTG/ABC/3TC |

All values at the time of cell sample collection

[a]*ABC* abacavir, *DDI* didanosine, *NVP* nevirapine, *3TC* lamivudine, *TDF* tenofovir, *D4T* stavudine, *IDV* indinavir, *RTV* ritonavir, *LPV* lopinavir, *AZT* zidovudine, *EFV* efavirenz, *FTC* emtricitabine, *DTG* dolutegravir, *MRV* maraviroc, *RIL* rilpivirine, *ELV* elvitegravir, *COBI* cobicistat, *TAF* tenofovir alafenamide fumarate, *DRV* darunavir, *VL* HIV-1 plasma viral load, *NA* not available

CD4 T-cell activation as measured by HLA-DR and CD69 expression (Fig. 4e), further indicating that CD32 expression is a marker of T-cell activation.

**CD32 expression does not correlate with integrated HIV-1 DNA.** To confirm the role of CD32 in HIV-1 infection, purified CD4+ T cells from 10 HIV+ individuals under ART were sorted using CD32 expression, and integrated provirus DNA was measured using qPCR. In 6 HIV+ individuals, integrated proviral DNA/cell was more prevalent in CD32− than CD32+ cells (Fig. 5a). However, there were no significant differences in the mean HIV-1 integrated provirus DNA/cell between the sorted populations (Fig. 5b). The mean contribution of HIV integrated provirus DNA was significantly higher in CD32− than in CD32+ cells ($p = 0.017$, Fig. 5c), indicating that the vast majority of infected CD4+ T cells appear to be CD32-negative.

**CD32 does not mark a replication-competent HIV-1 reservoir.** To understand the significance and contribution of CD32+ cells in the maintenance of the HIV-1 reservoir, equal numbers of sorted CD32+ or CD32− CD4+ T cells from a subset of HIV-1+ participants were used to perform a quantitative viral outgrowth assay based on an ultrasensitive co-culture with stimulated donor cells for 21 days[35]. Co-culture supernatants were titrated in CD4+ TZM-bl cells to evaluate the replication competence of the amplified virus, which was measured as luciferase production. In this model, released virus from CD32+ or CD32− CD4+ T cells should be competent enough to enter target cells and at least mediate Tat-dependent luciferase expression. There were no significant differences (Student's t-test, $p = 0.95$) in the mean maximum likelihood estimate of infection frequency (in infectious units per million, IUPM) between CD32− and CD32+ cell cultures (Table 2). Excluding participants P15 and P22, in which virus outgrowth could not be determined for CD32+ cells, the estimated IUPM did not fall within the same order of magnitude for CD32− and CD32+ cells in only two comparisons: P5 and P23. This finding suggests that the virus that emerges after the stimulation and co-culture of most of the HIV+ participant cells was similarly infectious regardless of CD32 expression.

### Discussion

HIV is characterized by chronic immune activation that drives viral replication and persistence[36,37]. The best characterized and the most likely mechanism for HIV persistence is the generation and maintenance of a "silent" reservoir of proviruses in resting memory CD4+ T cells of HIV+ individuals. Latently HIV-1-infected CD4+ T lymphocytes are a priori indistinguishable from uninfected lymphocytes[36], and they persist even during effective ART. Importantly, following activation, these cells can reactivate and start producing infectious virions, thus representing one of the main barriers to HIV-1 eradication.

Here we evaluated the role of CD32 expression in HIV-1 infection. CD32 is a cell surface protein recently proposed to be a marker of the CD4 T cell HIV reservoir[12]. We found that CD32 expression is strongly associated with CD4+ T cells that co-express the activation markers HLA-DR and/or CD69. Exogenous activation of purified CD4+ T cells with PHA, anti-CD3/CD28 mAbs or IL-7-induced CD32 cell surface expression correlating with cell proliferation (Ki67). We excluded the possibility of contaminating CD32+ monocytes (CD14+) or B cells (CD19+) and concluded that CD32 expression is a marker of activation in a subset of CD3+ CD4+ T cells, as recently suggested[38]. Evaluation of cells from HIV+ individuals showed similar results, with ~90% of CD32+ CD4+ T cells co-expressing

CD4+ T cells (Fig. 3a). The effect was dependent on the multiplicity of infection used (Fig. 3b) and was inhibited by efavirenz (Fig. 3c), indicating that the effect was dependent on productive HIV-1 replication. However, only a small fraction of HIV-1+ cells were CD32+ (Fig. 3d), and the ratio of HIV+ infected to uninfected cells did not significantly change depending on CD32 expression (18% vs. 16%; Fig. 3a, right panel). This finding indicated that CD32+ cells were not preferentially infected compared with HLA-DR+ cells. These results are in line with the observation that most CD32+ cells are activated (HLA-DR+ and/or CD69+), but not all activated cells are CD32+ (Fig. 1).

An alternative strategy to evaluate HIV-1 infection and latency in CD4+ resting cells is to allow purified resting cells to be permissive for HIV-1 infection after degrading SAMHD1[12], a restriction factor[29,30] that is active in resting cells[31,32]. Here we recapitulated this strategy with an HIV-1 NL4-3 virus modified to incorporate Vpx into HIV-1 virions (HIV-1* Vpx GFP)[31,33,34] and effectively infect resting (IL-2 only) purified CD4+ T cells (Fig. 4a). Infection with HIV-1* Vpx GFP-induced CD32 expression. The induction was dependent on the viral input (Fig. 4b) and blocked by efavirenz (Fig. 4c). After a 48-h incubation, cells were sorted using CD32 expression. The contribution of proviral DNA in CD32+ cells was evaluated by measuring integrated provirus DNA. We found more integrated DNA copies in the CD32− compartment in 4 out of 5 uninfected donor cells tested (Fig. 4d). The preferential infection of CD32+ cells in one donor (D2) was associated with significantly higher

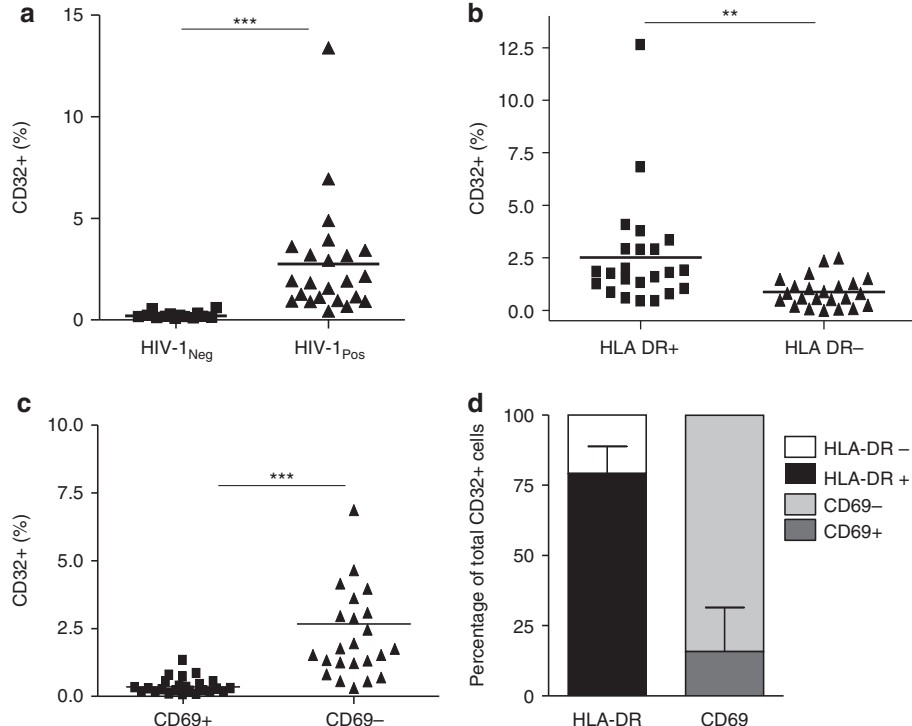

**Fig. 2** CD32 expression is associated to HLA-DR in CD4+ T cells from HIV-1+ individuals. **a** Expression of cell surface CD32 measured by flow cytometry in CD4+ T cells from uninfected donors (HIV-1$_{Neg}$) and HIV-1+ individuals (HIV-1$_{Pos}$). **b** and **c** Percentage of CD32+ cells in HIV-1-infected individuals co-expressing HLA-DR (**b**) or CD69 (**c**). CD32 cell surface expression was measured in 23 HIV-1+ individuals by flow cytometry in combination with the HLA-DR (left panel) or CD69 (right panel) activation markers. **d** Contribution of the HLA-DR and CD69 cells to the CD32 compartment in CD4+ T cells from HIV-1+ individuals. The data represent the mean ± SD from uninfected donors (N = 14) and HIV-1+ individuals (N = 23). Student's t-test, **p < 0.01, ***p < 0.001. The gating strategy used to identify CD32+ cells is shown in Supplementary Fig. 4. CD32 expression for each HIV+ individual is shown in Supplementary Fig. 5

the activation marker HLA-DR. Taking into account the fact that the HIV-1 reservoir is thought to be composed of resting CD4+ T cells, the elevated expression of activation markers and the correlation with cell proliferation on CD32+ cells observed here is somewhat counterintuitive.

Activation of CD4+ T cells is associated with HIV-1 pathogenesis and the establishment of an HIV-1 reservoir. Indeed, the HIV-1 reservoir is thought to form from cells that are infected while activated before returning to a resting state. There is a consistent association between the frequency of CD4+ and CD8+ T cells expressing HLA-DR and the frequency of resting CD4+ T cells containing HIV DNA. HLA-DR is also associated with productive HIV infection[39]. CD32 expression, like HLA-DR upregulation during T-cell activation, may identify a subset of activated CD4+ T cells that are susceptible to HIV infection. When challenged ex vivo with HIV-1, we found that CD32 expression was induced by infection itself similar to cell activation markers. This finding further indicates that CD32 expression is simply a consequence of T-cell activation induced either by exogenous stimuli or HIV-1 infection. However, we found no significant differences between the ratio of infected (GFP+) cells in CD32+ compared with CD32− cells, indicating that CD32 is not a preferential marker for infection, even though the majority of CD32+ cells also co-express the activation marker HLA-DR. Similar results were found in resting cells. That is, increased CD32 expression after infection of CD4+ resting T cells with a modified HIV-1 circumvents SAMHD1-induced restriction[12,31,40] allowing productive infection. Indeed, proviral DNA in in vitro infected resting cells was preferentially found in the CD32− cells, except for one donor (n = 5) in which T-cell activation (HLA-DR and CD69 expression) was significantly

higher. This finding further demonstrated the link between cell activation and CD32 expression.

Taken together, the correlation between cell activation and CD32 expression may suggest that CD32+ CD4 T cells have a history of activation consistent with the current understanding of how the reservoir develops and is maintained. Specifically, HIV-1-infected resting memory CD4+ T lymphocytes mirror a post-activation state, in which infection and subsequent return to a lower activation level occurred[41]. Alternatively, CD32+ cells may reflect antigen-independent homeostatic proliferation. For example, anti-CD3 mAbs were observed to induce T-cell proliferation mediated by CD32[42]. We show that IL-7-induced proliferation caused a moderate increase in CD32 expression. The hypothesis that CD32 cell surface expression would allow for the selective recognition of an HIV-1 reservoir contradicts the paradigm of the "undistinguishable phenotype of latently infected cells"; however, the data could be interpreted as indicating that latently infected cells would no longer be resting as HIV-1 infection leads to expression of an immune modulator (CD32) and likely 103 other genes, as suggested by Descours et al.[12].

Our data also challenge the robustness of CD32 as a marker of an HIV-1 reservoir. We found that in 6 out of 10 HIV+ individuals, the absolute contribution to the CD4+ T cell HIV-1 reservoir was higher in CD32− CD4+ T cells. These results are in line with the raw data of Descours et al.[12] (provided as a Supplementary Excel file to Fig. 3c in their publication) in which the absolute contribution of HIV-proviral DNA copies by CD32+ cells was higher in only 5 of 9 HIV+ individuals, and in one, the contribution was comparable between CD32− and CD32+ cells. Taken together, these data indicate that CD32+ cells are not a preferential HIV reservoir in all HIV+ individuals.

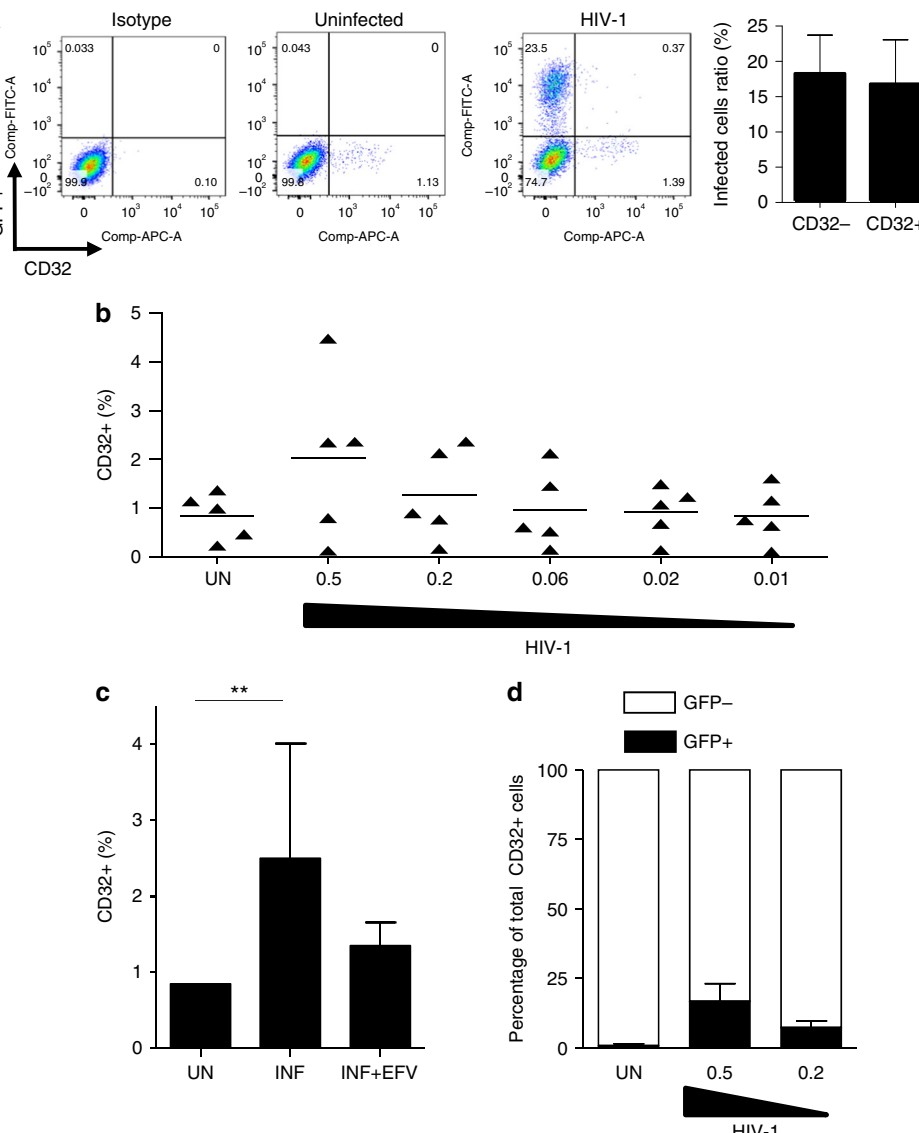

**Fig. 3** CD32 is upregulated after productive HIV-1 infection. **a** CD32 expression in PHA/IL-2 stimulated CD4+ T cells infected with HIV-1 (a representative donor out of 5 is shown). The right panel indicates the ratio of HIV+ (GFP+) to CD32− or to CD32+ cells (N = 5). **b** Percentage of CD32 cell surface expression measured by flow cytometry and infected with different multiplicities of infection of HIV-1 NL4-3 or uninfected (UN). **c** Upregulation of CD32 expression after HIV-1 infection (INF) is reduced concomitant to blockade of HIV-1 infection with efavirenz (INF + EFV). **d** The percentage of HIV-1+ (GFP+) cells in the CD32+ compartment. The data represent the mean ± SD of five different donors

Grau-Exposito et al.[43] recently showed that HIV mRNA+ cells express CD32, and the authors suggested that CD32 expression is restricted to cells productively infected by HIV-1. Establishment of HIV-1 latency may be the consequence of infection in CD4+ T cells within a narrow window of time after activation[41]. Thus, CD32 expression may signal a transition state to or from a fully susceptible phenotype.

Intriguingly, the integrated virus in CD32+ cells appears to account for most, if not all, of the replication-competent HIV-1[12]. Previous studies identified latent replication-competent HIV-1 in different CD4+ T-cell subsets[44–46]. The proportion of intact proviruses was different, but shared an identical sequence across cell subsets[44]. These observations suggest maintenance of the viral reservoir by cellular proliferation, which is also associated with CD32 expression. Our results indicate that there are no significant differences between replication competence of viruses emerging from CD32− and CD32+ CD4 T cells, in line with the recent findings indicating that CD32 did not enrich for HIV

latently infected cells[47] and in contrasts with the results from Descours et al. The TZM-bl assay used in our study records virus that must enter target cells and induce Tat-dependent transcription to register a positive signal but may still be defective in numerous ways. Thus, the assay may be overestimating replication-competent HIV-1, explaining the higher IUPM values observed in our study than that recorded by other groups in quiescent, persistently infected CD4+ T cells. However, we compared the viral outgrowth of cultures with an equal cell number for CD32− and CD32+ cells, allowing for head-to-head comparisons between both cell types. Conversely, Descours et al. compared dilutions of 1 million to 1600 total CD4+ T cells and dilutions from 800 to 1 CD32+ cells. Differences in our cell culture conditions may also affect the experimental outcome.

Nevertheless, taken together, our results do not support the existence of a distinct population of latent provirus in CD32+ and CD32− cells; both cell types harbored similar amounts of integrated provirus with similar replication competence. A better

ARTICLE

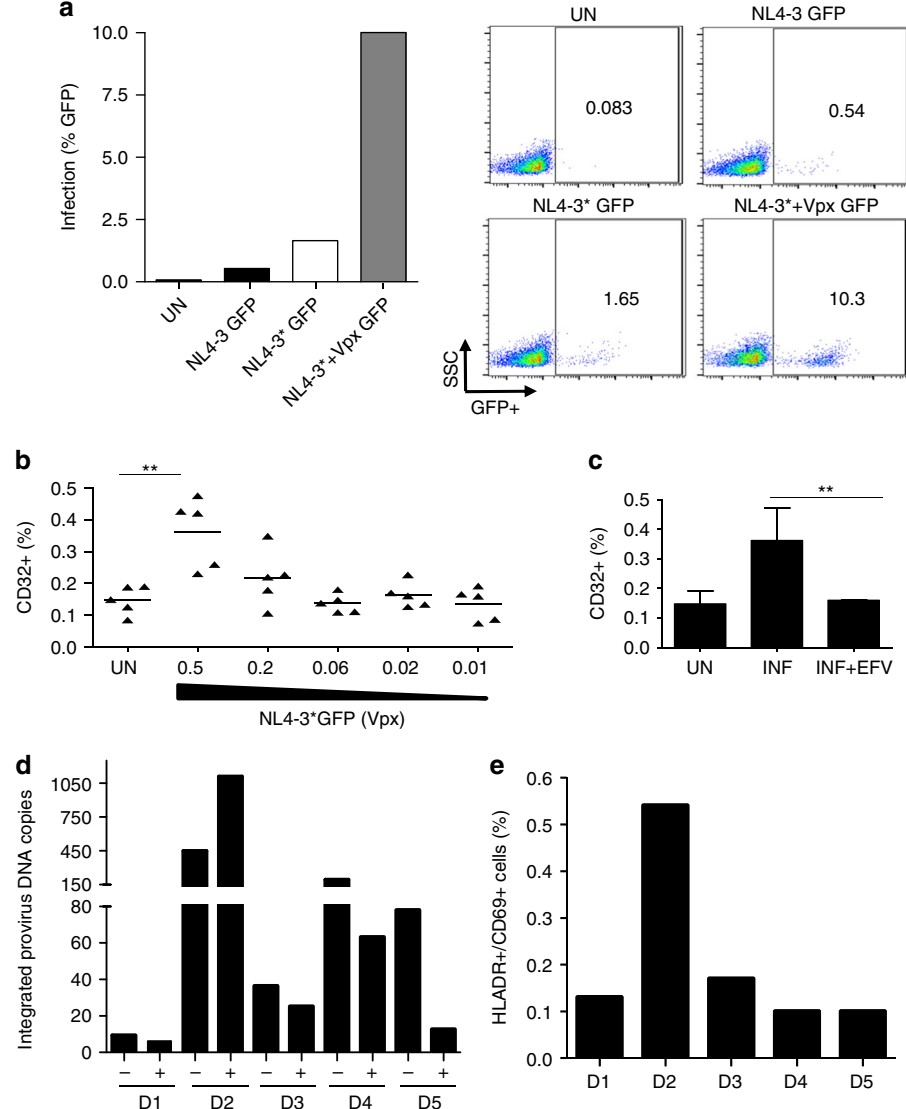

**Fig. 4** Contribution of HIV-1 proviral DNA in CD32+/CD4+ T cells from in vitro infections. **a** Infection of CD4+ T cells treated with IL-2 and infected with NL4-3GFP, NL4-3*GFP, and NL4-3*GFP carrying Vpx. The percentage of infection was evaluated using flow cytometry; representative dots are shown on the right. The data from a representative donor are shown. **b** Percentage of CD32 cell surface expression measured by flow cytometry and infected with different multiplicities of infection of HIV-1 NL4-3 carrying HIV-2 Vpx or uninfected (UN) ($n = 5$). Lines represent mean values. **c** Upregulation of CD32 expression after infection (INF) is reduced concomitant to blockade of HIV-1 infection with efavirenz (INF + EFV) The data represent the mean ± SD of five different donors. For (**b**) and (**c**) Student's $t$-test, **$p < 0.005$. **d** Integrated HIV-1 DNA copy number in sorted CD32+ and CD32− cells of five different donors infected with NL4-3*(Vpx). Measurement of integrated proviral DNA was performed by pre-amplifying an LTR DNA fragment with equal amount of genomic DNA input (100 ng) from the sorted CD32+ or CD32− population. Absolute quantification was obtained in a second amplification of HIV-LTR by qPCR. The data from each donor are shown. **e** Activation level of CD4+ T cells from five uninfected donors as measured as the expression of HLA-DR and CD69 cell surface markers by flow cytometry

characterization and immune phenotyping of the CD32+ cell subset is important to clearly determine the exact contribution of CD32-expressing cells to the replication-competent-latent reservoir.

Establishing a competent-latent reservoir requires intact retroviral integration into the genome and subsequent transcriptional silencing. Taking into account that CD32 expression is activation-dependent but may not represent a marker of latent infection, the CD32 contribution to the establishment of competent-latent reservoir could be explained if CD32 expression is triggered by a relatively late event in virus replication in cells that would become silent and leave the CD32 signature. Thus, the CD32 marker would only be expressed in cells with competent HIV-1. In contrast, integrated but not fully replicated virus would

not trigger the CD32 expression signal. However, HIV infection leads to the perturbation of cellular transcription in hundreds of genes induced by HIV-dependent transcriptional activation[48], and of course, CD32 expression could also be the result of this phenomenon.

In summary, our data suggest that CD32 expression represents a marker of activation in a subset of CD4+ T cells, rather than a marker of the HIV-1 reservoir. However, considering the observed association between immune activation and viral persistence suggests that these two phenomena may be reciprocally connected; a putative role of CD32 in such a scenario cannot be ruled out. Thus, the role of CD32 in establishing an HIV-1 latent reservoir still requires exploration and discussion because of its implications in designing therapeutic strategies for HIV.

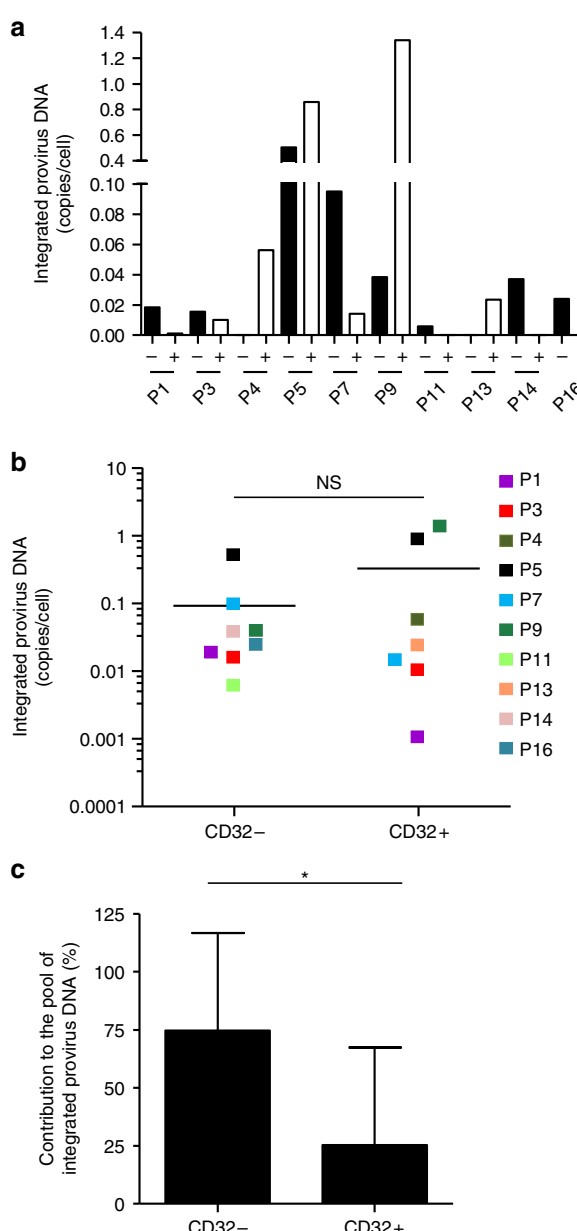

**Fig. 5** Contribution of HIV-1 proviral DNA in the CD32+/CD4+ T cells from HIV-1 individuals. **a** Quantification of HIV-1 integrated provirus HIV-1 DNA copies per cell in CD32− (black bars) and CD32+ (white bars) CD4 T cells from 10 ART-treated individuals. **b** Individual HIV-1 integrated provirus DNA copies per cell in CD32− and CD32+ CD4 T cells. The mean values are presented as horizontal lines. Each color represents values from the same HIV+ individual. NS not significant ($p = 0.3$). **c** Relative contribution of HIV-1 integrated provirus DNA in the CD32− and CD32+ CD4 T cell compartments from 10 ART-treated individuals. The mean values are presented as the percentage relative to the total number of integrated provirus DNA copies. Student's t-test, *$p < 0.05$

Confirmation of these findings could also redefine the concept of resting immune cells, not just for HIV.

## Methods

**Cells.** PBMCs were isolated from the buffy coats of uninfected blood donors. The buffy coats were purchased from the Catalan Banc de Sang i Teixits (http://www.bancsang.net/en/index.html; Barcelona, Spain). The buffy coats were anonymous and untraceable, and the only information provided was whether they had been tested for disease. Briefly, PBMCs were obtained using a Ficoll-Paque density gradient centrifugation and used for fresh purification of CD4+ T lymphocytes by

**Table 2 IUPM values for CD32− and CD32+ CD4+ T cells**

| Participant | CD32− CD4+ T cells | 95% CI | CD32+ CD4+ T cells | 95% CI |
|---|---|---|---|---|
| P1 | 2039 | 762–5455 | 45,425 | 12,356–166,993 |
| P2 | 5634 | 1266–20,245 | 8463 | 2452–23,579 |
| P3 | 203,973 | 72,119–576,893 | 274,243 | 103,634–725,716 |
| P5 | 6023 | 2684–13,514 | 10,414 | 2787–38,914 |
| P7 | 277 | 103–740 | 888 | 419–1882 |
| P9 | 10,207 | 2543–40,968 | 9049 | 2263–36,618 |
| P15 | 11,556 | 2879–46,380 | Und | — |
| P16 | 2253 | 1121–4523 | 2440 | 1209–4924 |
| P19 | 482 | 117–1978 | 189 | 26–1344 |
| P20 | 159,293 | 48,697–521,056 | 330,601 | 121,121–902,375 |
| P21 | 189 | 26–1360 | 524 | 120–2271 |
| P22 | 2835 | 654–12,279 | Und | — |
| P23 | 128,163 | 68,593–239,465 | 7856 | 4698–13,119 |
| Mean ± SD | 40,994 ± 71,810 | | 42,401 ± 69,174 | |

95% CI, lower bound and upper bound of 95% confidence interval, calculated according to Rosenbloom et al.[50]
IUPM infectious units per million cells, Und undetermined, SD standard deviation of the mean

the EasySep™ Human CD4+ T Cell Enrichment negative selection Kit (StemCell Technologies, catalog number 19052). Purity of the populations was confirmed using flow cytometry. Both isolated CD4+ T lymphocytes and total PBMCs were kept in complete RPMI 1640 (Thermofisher/Gibco) supplemented with 10% heat-inactivated fetal bovine serum (FBS; Thermofisher/Gibco), penicillin and strepto-mycin (Thermofisher/Gibco), with IL-2 alone (3 ng/ml, Sigma-Aldrich, catalog number 011011456001) or IL-2 and one of the following stimuli PHA (4 μg/ml; Sigma-Aldrich, catalog number L1668), anti-CD3 and anti-CD28 antibodies (Immunocult™, StemCell Technologies, catalog number 10991) or IL-7 (5 ng/ml, Peprotech, catalog number 200-07) when appropriate. ACH2 and CD4+ TZM-bl cell lines were obtained from the AIDS Reagent Program, National Institutes of Health, Bethesda, MD, USA (catalog number 349 and 103, respectively). ACH2 and CD4+ TZM-bl cells were grown in RPMI 1640 or DMEM medium, respectively, supplemented with 10% of heat-inactivated fetal calf serum (FCS, Gibco, Life Technologies, Madrid, Spain) and antibiotics 100 U/ml penicillin, 100 μg/ml streptomycin (Life Technologies) and maintained at 37 °C in a 5% $CO_2$ incubator.

**HIV+ individuals.** All participants in the study provided informed consent, and the work was approved by the Scientific Committee of Fundació IrsiCaixa and the Ethics Committee of Hospital Germans Trias i Pujol (Ref. CEI PI-18–021). All methods were performed in accordance with relevant guidelines and regulations and the ethical principles suggested in the Declaration of Helsinki. Subject samples were included if the individuals were older than 18 years old, had chronic HIV-1 infection and had previously been on highly active ART for >1 year. HIV-RNA levels were <400 copies/ml during at least 1 year and <50 copies/ml at study entry. Immunological and virological characteristics from all participants are found in Table 1. All participants were males. Frozen PBMCs (isolated as described above and stored at −80 °C) or cells isolated from fresh peripheral blood from HIV+ individuals visiting our clinic were used.

**Virus and virus infections.** The NL4-3-GFP wild type or modified to bind Vpx (NL4-3*GFP) were kindly provided by Dr. O. T. Keppler (Max von Pettenkofer Institute, Ludwig-Maximilians-Universität Munich, Germany). Wild-type NL4-3GFP or NL4-3*GFP[31,34] were co-transfected with the Vpx expression construct SIVmac239- into HEK293T cells to produce viral stocks. Three days after transfection, the supernatants were collected, filtered, and concentrated using Lenti-X concentrator (Clontech, Catalog number 631232) and stored at −80 °C. Infections were performed using spinoculation (1200 g, 2 h at 37 °C) using $0.25 \times 10^6$ cells/well. After spinoculation, cells were kept in the incubator for 72 h prior to analysis by flow cytometry. Efavirenz (Sigma-Aldrich, Catalog number SML0536) was added when appropriate.

**mRNA quantification.** For relative mRNA quantification, RNA was extracted using the NucleoSpin RNA II kit (Magerey-Nagel, Catalog number 740955) as recommended by the manufacturer, including the DNase I treatment step. Reverse transcription was performed using the PrimeScript™ RT-PCR Kit (Takara, Catalog number RR036A). The relative level of FCGR2a gene transcription was measured using two-step quantitative RT-PCR and normalized to GAPDH mRNA expression using the ΔΔCt method[49]. Primers and DNA probes were purchased from Life Technologies (TaqMan gene expression assays).

**Integrated HIV-1 provirus DNA quantification**. DNA was extracted using the DNA Quick extraction kit from Epicentre following the manufacturer's instructions. For integrated provirus DNA quantification, an LTR pre-amplification was performed to assure amplification of integrated HIV-1 only (forward 5′-GCCTCCCAAAGTGCTGGGATTACAG-3′ or 5′-TGGCAGAACTACACA CCAGG-3′; reverse 5′-TTGCCCATACTATATGTTTTAA-3′) followed by quantitative PCR amplification of an internal LTR fragment using the following primers and probe: forward 5′-GACGCAGGACTCGGCTTG-3′, reverse 5′-ACTGACGCT CTCGCACCC-3′, and probe FAM 5′-TTTGGCGTACTCACCAGTCGCCG-3′ TAMRA. Absolute quantification was obtained by extrapolating from Ct data with a standard curve performed in parallel with a series of samples of known HIV-1 copy number, based on the ACH2 cell line.

**Flow cytometry**. Cells were labeled with the following antibodies: anti-human CD3 PerCP (4 μl, BD catalog number 340663); anti-human CD8 BV510 (3 μl, BD catalog number 740175); anti-human CD69-BV421 (3 μl, BD catalog number 562884); anti-HLA DR PeCy7 (3 μl, BD catalog number 335795); anti-CD14-FITC (3 μl, catalog number 347493) or anti-CD14-PE (3 μl, BD catalog number 562334); and anti-CD19-FITC (3 μl, BD catalog number 347543); and anti-human CD32-APC (4 μl) (Sony Biotech, catalog number 2116040) for the characterization of donor PBMCs and HIV+ individuals. Cells were incubated for 40 min at room temperature in the dark. Cells were washed with phosphate-buffered saline (PBS) and fixed with 1% formaldehyde (FA) prior to cytometer acquisition. For the determination of CD32 expression levels, an APC mouse IgG2b (4 μl) (Sony Biotech, catalog number 2601600) was included as isotype control, using a threshold value ≤0.1 in all cases. Flow cytometry was performed in a FACS LSRII flow cytometer (BD Biosciences). The data were analyzed using the FlowJo software (Tree Star Inc., Ashland, OR).

For cell-sorting experiments, CD4+ T cells were labeled as described above for 40 min at 37 °C and kept in PBS supplemented with 1% FBS. The different CD4+ T cell subpopulations were identified by FACS, and the CD3+ CD8− CD14− population was sorted into the CD32+ or CD32− fractions using FACSAria II (BD Biosciences). For intracellular Ki-67 staining, cells were fixed for 3 min with fixation buffer (FIX & PERM; Life Technologies Life technologies, catalog number GAS004) before adding precooled 50% methanol for 10 min at 4 °C. Cells were washed in PBS with 5% FBS and incubated for 30 min with the Ki-67 FITC Ab (1:10; clone B56; BD Biosciences, catalog number 556026) diluted in permeabilization buffer.

**Quantitative viral outgrowth assay**. An ultrasensitive co-culture assay was applied to sorted CD32+ or CD32− CD4+ T cells isolated from a subset of HIV+ individuals on ART[35]. Briefly, purified cells (500–20,000 CD4+ cells) were stimulated with a pool of allogeneic irradiated PBMCs at a ratio of 1:5 with allogeneic PBMCs in 96-well plates in the presence of PHA (1 μg/ml) and IL-2 (100 U/ml) for 72 h and co-cultured for 7 days with a pool of stimulated CD8-depleted PBMCs from 3 HIV-seronegative donors. To maximize viral outgrowth during the following 2 weeks, the co-cultures were fed once a week with fresh medium and once a week with a pool of stimulated CD4+ cells from three HIV-seronegative donors. After 21 days in culture, the supernatants were assayed in CD4+ TZM-bl cells (NIH AIDS Reagent Program, catalog number 8129)[35], and the number of infectious units per million of cells (IUPM) were calculated according to Rosenbloom et al.[50] with the use of the IUPM Stats v1.0 Infection Frequency Calculator (http://silicianolab.johnshopkins.edu). The use of reporter CD4+ TZM-bl cells has been shown to have a 1000-fold increase in sensitivity and helped demonstrate that the size of the inducible latent HIV-1 reservoir in aviremic participants on therapy may be ~70-fold larger than previous estimates[51].

**Statistical analyses**. All experimental data are presented as the mean ± standard deviation (SD) calculated using Student's t-test with the GraphPad PRISM software (GraphPad Software, San Diego, CA, USA).

**Data availability**. All relevant data are available from the authors.

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

## Acknowledgements

We thank all HIV+ individuals participating in the study and the National Institutes of Health (NIH) AIDS Reagent Program for reagents. This work was supported in part by the Spanish Ministerio de Economía y Competitividad (MINECO) projects BFU2015–63800-R, FIS PI15/00492, PI16/00103, and PI17/00624.

## Author contributions

J.A.E. and E.R.-M. conceived the study. R.B., E.B., M.C., E.R.-M., M.P., E.G.-V., and J.G.P. designed and conducted the experiments. J.P. processed samples and analyzed the data. J.A.E., M.A.M., and B.C. recruited participants and analyzed the data. All authors read and approved the final manuscript.

## Additional information

**Competing interests:** The authors declare no competing interests.

