## [Peer Review File · Nature Communications]

Reviewers' comments:

Reviewer #1 (Remarks to the Author):

Badia et al. evaluated the impact of T cell activation on CD32 expression. They observed that different stimuli (TCR triggering, IL-2 and IL-7) increase IL-32 expression in CD4+ T cells and that these cells frequently co-expressed HLA-DR. Importantly, higher frequencies of CD32+ CD4+ T cells were measured in HIV-infected individuals compared to uninfected controls, and these cells frequently expressed HLA-DR but not CD69. In vitro infection showed that only a small fraction of productively infected cells expressed CD32. In addition, HIV DNA contents were similar in the CD32- and CD32+ subsets. Using samples from suppressed individuals, the authors show that CD32 does not enrich in CD4+ T cells harboring HIV DNA and that the majority of infected cells do not express CD32.

Although the results of this paper are potentially interesting, there are several important issues that preclude its publication. In Figures 1 and 2, negative controls are lacking and several panels are not described in the main text. Statistics should be entirely revised throughout the manuscript (many panels do not include p values). The nature of the PCR assay (total or integrated) is critical and should be better explained. Several data are hard to understand, since they are presented in an unconventional manner.

1. There is a misconception of the results from Descours et al. Unlike what the authors state in the first line of the abstract, CD32 has not been shown to mark latently infected cells in vivo (although it was first identified in a latency model). Rather, CD32 identifies HIV-infected cells in ART-suppressed individuals regardless of their activation and latency status. This difference is important in the context of the present study. The manuscript should be corrected accordingly and more details on the original study should be included. For instance, it would be important to mention that the differentially expressed genes (line 37) were identified in vitro.
2. Statistics are lacking in many figures (1a, 1c, 1e, 2a-c, 3c, 4b, 4d, 5a, 5d, 5e, 6a, 6c, 6d). Appropriate statistical tests should be used and justified and p values clearly indicated in the figures.
3. Line 70: 4 different stimulations are tested but only 3 are presented in Figure 1a. The IL-2+IL-7 results should be presented and IL-2 should be added to CD3/28 on the figure, for consistency with the text. The condition presented in Figure 2 (IL-7 alone?) is not described in line 70. I would recommend to merge Figure 1 and 2 and use similar ways to represent the data in all stimulation conditions. Figure 2 is poorly described and panel 2d (Ki67) not even mentioned in the main text.
4. In figure 1c, unstimulated cells (negative control) should be shown.
5. Results from Figure 1e-f suggest that CD32, HLA-DR and CD69 may be expressed through different signaling pathways since they do not overlap entirely. Whereas it looks like TCR engagement (with CD3 Ab) induces concomitant expression of HLA-DR and CD32, CD69 is rarely co-expressed by these cells. Knowing that the kinetic of expression of these markers differ after activation, how these results should be interpreted? Also, expression of HLA-DR and CD69 on non-stimulated CD32+ cells should be shown.
6. The Y axis of Figure 3c should be modified, as this figure does not represent the frequency of HLA-DR or CD69 cells expressing CD32, but rather the frequency of CD32+ cells expressing HLA-DR or CD69. Rather than correlations (line 69), these data show associations.
7. It is unclear if the assay used by the authors quantified total or integrated HIV DNA. The methods section (line 450) describes an assay that measures all genomes (integrated or not), which is in conflict with the title of the section, the figures and the main text. Proviral DNA should be used for integrated genomes only. This should be clarified. Of note, the original study by Descours et al. suggested that the integration step may be required for CD32 expression. Therefore, the use of an integrated assay in these in vitro experiments is critical.
8. Figure 6 is also problematic on multiple levels. Panels a and b show the number of HIV copies per cell (like in Descours et al), which was most probably calculated by making the ratio between the HIV DNA copies (again, it is unclear if these are integrated or not) and the number of cells. I

do not understand the added value of panel c (absolute number of HIV DNA copies), since it is obvious that these numbers will depend on the number of cell used in this assay, which are unlikely to be similar between the subsets and the samples. Finally, panel d does not show a "contribution" (which should be represented as a percentage). Rather, the authors should represent the relative contributions of the CD32- and CD32+ subsets to the entire pool of infected cells (100%).

9. Line 187: I think there is a confusion between the frequency of infected cells (which is a ratio) and the contribution of a subset (which is a percentage).

10. As acknowledged by the authors, the manuscript does not include measurements of the replication competent reservoir, which limits the impact of the findings.

11. Several terms used in this paper should be replaced. Community representative prefer to no longer use the word "patient", particularly for HIV infected people on ART. Rather, "HIV infected individuals" or "participants" should be used. Similarly, "healthy donors" should be modified for control donors or uninfected controls.

Reviewer #2 (Remarks to the Author):

The authors explore the validity of Descours et al. on the ability of CD32A to mark the latent HIV reservoir. Their findings generally contradict those previously published, and they speculate on some of the possible reasons for their different observations. It is important to share this information.

The work is convincing, as far as it goes, and although the question of the frequency of replication-competent HIV in the CD32 + and negative populations is unaddressed, it was not really sufficiently addressed in the Descours work as well.

While the discussion was readable, some of the writing style was awkward and its meaning might have been clarified. Light editing for style might be helpful.

Reviewer #3 (Remarks to the Author):

The manuscript by Badia et al seeks to validate the use of CD32 expression on CD4+ T cells as a marker of cells that carry latent HIV originally originally published as an observation by Descours et al. As the author's discuss a cell surface expressed protein that can be readily used to identify the small number of resting CD4+ T cells that comprise the HIV latent reservoir would extremely useful for the study and targeting of latently infected cells.

Their data show that a subset of resting CD4+ T cells express CD32 following stimulation using a number of polyclonal activation stimuli, in comparison to HLA-DR and or CD69 expression this is a minor population cells.

The authors also looked for CD32 expression on CD4+ T cells from HIV infected individuals that had good virus control (<50 copies/ml), this data showed that CD32 expression was significantly higher in HIV+ individuals and also associated with HLA-DR expression.

Activated CD4 T cells infected with HIV induced CD32 expression which was partially blocked by a HIV RT inhibitor. However CD32- HLA-DR+ cells were also infected. Utilizing a similar approach to Descours et al a system that allowed HIV-1 infection without activating CD4+ T cells was used and showed that CD32 was induced upon infection in agreement with published observation. These experiments also showed that in most donors used the proviral DNA was predominantly in CD32- T cells.

10 HIV+ individuals were used to derive CD32+ and - CD4 T cells and qPCR used to measure proviral load, this recapitulates the experiment from Descours et al fig4b. The result from this analysis showed no statistical difference in proviral load between CD32- and CD32+ and only 2/10 individuals had a higher load in CD32+ T cells. They conclude that the majority of infected CD4 T cells are CD32-.

This group were unable to perform viral out growth assays to be able to compare if this proviral population could give rise to reactivatable replication competent HIV.

This group conclude that their data suggest that CD32 expression identify activated CD4 T cells, but is not a marker of the latent reservoir. They discuss that their results show in 6/10 HIV+ donors tested that proviral copies/ cell were higher in CD32- T cells and that this was comparable to Descours et al raw data in supplemental data (5/9 HIV+ donors), however it is unclear what specific reinterpretation of raw data that is being performed here? This requires clarification.

The discussion is not clear, as to why the results from this publication are different from Descours et al., this clearly needs to be resolved. The ability of the latent reservoir to reactivate and produce replication competent virus is clearly an important attribute over and above just proviral load. Performing these experiments would be highly desirable in this situation and these experiments should be performed, it needs to be addressed if CD32+ T cells contribute the majority of replication competent HIV or if again the CD32- T cell population is equally important in contributing to the reactivatable replication competent reservoir ?

Minor points the English in the paper could benefit from editing to help with the flow and readability.

Response to reviewers.

Reviewer #1:

Badia et al. evaluated the impact of T cell activation on CD32 expression. They observed that different stimuli (TCR triggering, IL-2 and IL-7) increase IL-32 expression in CD4+ T cells and that these cells frequently co-expressed HLA-DR. Importantly, higher frequencies of CD32+ CD4+ T cells were measured in HIV-infected individuals compared to uninfected controls, and these cells frequently expressed HLA-DR but not CD69. In vitro infection showed that only a small fraction of productively infected cells expressed CD32. In addition, HIV DNA contents were similar in the CD32- and CD32+ subsets. Using samples from suppressed individuals, the authors show that CD32 does not enrich in CD4+ T cells harboring HIV DNA and that the majority of infected cells do not express CD32.

Although the results of this paper are potentially interesting, there are several important issues that preclude its publication. In Figures 1 and 2, negative controls are lacking and several panels are not described in the main text. Statistics should be entirely revised throughout the manuscript (many panels do not include p values).

Response: Figures 1 and 2 have been revised. All panels are now clearly mentioned in the text and statistics are provided.

We have merged old Figures 1 and 2 into the new Figure 1, included data of cells without any stimulus as control. Results and conclusions have not changed: T cell activation induced CD32 expression in CD4+ T cells.

The nature of the PCR assay (total or integrated) is critical and should be better explained. Several data are hard to understand, since they are presented in an unconventional manner.

Response: The PCR assay measures integrated DNA and is now clearly explained in Methods. We have revised all figures and figure legends to clarify the presentation of data.

1. There is a misconception of the results from Descours et al. Unlike what the authors state in the first line of the abstract, CD32 has not been shown to mark latently infected cells in vivo (although it was first identified in a latency model). Rather, CD32 identifies HIV-infected cells in ART-suppressed individuals regardless of their activation and latency status. This difference is important in the context of the present study. The manuscript should be corrected accordingly and more details on the original study should be included. For instance, it would be important to mention that the differentially expressed genes (line 37) were identified in vitro.

Response: We disagree. As mentioned by the reviewer, Descours et al first identify CD32 in vitro in a latency model and only when referring to HIV+ individuals they describe CD32 as “marker of CD4 T-cell HIV reservoir...” Indeed, the HIV reservoir definition, i.e. cells (CD4 T cells or others) in which a replication-competent form of

HIV persists, entails the existence of viral latency. This is clearly stated in the News & Views article that accompanies the original Nature paper: “Descours et al. have identified just such a marker for about half of the latently infected CD4 T cells in the blood”. Also, Pillai and Deeks (Trend in Immunology, 2017) comment: “Descours et al. identifies CD32a as a marker of latently infected T cells”. And even in Descours et al the following is mentioned: “These results validate CD32a as a cell surface marker of CD4 T cell HIV reservoir in HIV-infected virally suppressed participants”. Thus, it is obvious that Descours et al suggested the identification of a latent reservoir marked by CD32, an issue that needs clear-cut clarification because the relevance of their study, if any, lays in the identification of a latent reservoir through CD32 expression. Nevertheless, we have modified our text to emphasize that CD32 marks a reservoir, but we defined such reservoir as that of latently infected cells. This clarification is of importance to adequately value our work and that of Descours et al.

2. Statistics are lacking in many figures (1a, 1c, 1e, 2a-c, 3c, 4b, 4d, 5a, 5d, 5e, 6a, 6c, 6d). Appropriate statistical tests should be used and justified and p values clearly indicated in the figures.

Response: Done

3. Line 70: 4 different stimulations are tested but only 3 are presented in Figure 1a. The IL-2+IL-7 results should be presented and IL-2 should be added to CD3/28 on the figure, for consistency with the text. The condition presented in Figure 2 (IL-7 alone?) is not described in line 70. I would recommend to merge Figure 1 and 2 and use similar ways to represent the data in all stimulation conditions. Figure 2 is poorly described and panel 2d (Ki67) not even mentioned in the main text.

Response: We have now merged Fig. 1 and 2, compounded the description of results and mentioned all figures in the text.

4. In figure 1c, unstimulated cells (negative control) should be shown.

Response: done.

5. Results from Figure 1e-f suggest that CD32, HLA-DR and CD69 may be expressed through different signaling pathways since they do not overlap entirely. Whereas it looks like TCR engagement (with CD3 Ab) induces concomitant expression of HLA-DR and CD32, CD69 is rarely co-expressed by these cells. Knowing that the kinetic of expression of these markers differ after activation, how these results should be interpreted? Also, expression of HLA-DR and CD69 on non-stimulated CD32+ cells should be shown.

Response: Taken together, our results clearly indicate that CD32 expression occurs in cells with a degree of activation, clearly described by HLA-DR expression and somewhat by CD69. It is out of the scope of this manuscript to disentangle the complex mechanism of early and late expression of T cells markers of activation. In

turn, we definitively show that CD32 is coexpressed with T cell activation markers. A sentence has been included in the Discussion to emphasize this issue. Moreover, as mentioned above, unstimulated cells have been included.

6. The Y axis of Figure 3c should be modified, as this figure does not represent the frequency of HLA-DR or CD69 cells expressing CD32, but rather the frequency of CD32+ cells expressing HLA-DR or CD69. Rather than correlations (line 69), these data show associations.

Response: done.

7. It is unclear if the assay used by the authors quantified total or integrated HIV DNA. The methods section (line 450) describes an assay that measures all genomes (integrated or not), which is in conflict with the title of the section, the figures and the main text. Proviral DNA should be used for integrated genomes only. This should be clarified. Of note, the original study by Descours et al. suggested that the integration step may be required for CD32 expression. Therefore, the use of an integrated assay in these in vitro experiments is critical.

Response: We used a method that evaluates integrated DNA. Methods section has been clarified.

8. Figure 6 is also problematic on multiple levels. Panels a and b show the number of HIV copies per cell (like in Descours et al), which was most probably calculated by making the ratio between the HIV DNA copies (again, it is unclear if these are integrated or not) and the number of cells. I do not understand the added value of panel c (absolute number of HIV DNA copies), since it is obvious that these numbers will depend on the number of cell used in this assay, which are unlikely to be similar between the subsets and the samples. Finally, panel d does not show a "contribution" (which should be represented as a percentage). Rather, the authors should represent the relative contributions of the CD32- and CD32+ subsets to the entire pool of infected cells (100%).

Response: We have eliminated panel c. Panel d (new panel c) has been modified to show the relative contributions of the CD32- and CD32+ subsets as a percentage. The Y axis has been modified to clarify that integrated DNA was measured.

9. Line 187: I think there is a confusion between the frequency of infected cells (which is a ratio) and the contribution of a subset (which is a percentage).

Response: corrected.

10. As acknowledged by the authors, the manuscript does not include measurements of the replication competent reservoir, which limits the impact of the findings.

Response: We have now included the evaluation of replication-competent HIV-1 as

requested in new Table 2.

11. Several terms used in this paper should be replaced. Community representative prefer to no longer use the word "patient", particularly for HIV infected people on ART. Rather, "HIV infected individuals" or "participants" should be used. Similarly, "healthy donors" should be modified for control donors or uninfected controls.

Response: We agree. The manuscript has been modified accordingly.

Reviewer #2 (Remarks to the Author):

The authors explore the validity of Descours et al. on the ability of CD32A to mark the latent HIV reservoir. Their findings generally contradict those previously published, and they speculate on some of the possible reasons for their different observations. It is important to share this information.

Response: We agree and thank the reviewer for considering our work relevant.

The work is convincing, as far as it goes, and although the question of the frequency of replication-competent HIV in the CD32 + and negative populations is unaddressed, it was not really sufficiently addressed in the Descours work as well.

Response: We have now included the evaluation of replication-competent HIV as requested in new Table 2.

While the discussion was readable, some of the writing style was awkward and its meaning might have been clarified. Light editing for style might be helpful.

Response: We have carefully revised our manuscript and the text has been edited by the Springer Nature Editing service.

Reviewer #3 (Remarks to the Author):

The manuscript by Badia et al seeks to validate the use of CD32 expression on CD4+ T cells as a marker of cells that carry latent HIV originally originally published as an observation by Descours et al. As the author's discuss a cell surface expressed protein that can be readily used to identify the small number of resting CD4+ T cells that comprise the HIV latent reservoir would extremely useful for the study and targeting of latently infected cells.

Their data show that a subset of resting CD4+ T cells express CD32 following stimulation using a number of polyclonal activation stimuli, in comparison to HLA-DR and or CD69 expression this is a minor population cells.

The authors also looked for CD32 expression on CD4+ T cells from HIV infected individuals that had good virus control (<50 copies/ml), this data showed that CD32 expression was significantly higher in HIV+ individuals and also associated with HLA-DR

expression.

Activated CD4 T cells infected with HIV induced CD32 expression which was partially blocked by a HIV RT inhibitor. However CD32⁻ HLA-DR⁺ cells were also infected. Utilizing a similar approach to Descours et al a system that allowed HIV-1 infection without activating CD4⁺ T cells was used and showed that CD32 was induced upon infection in agreement with published observation. These experiments also showed that in most donors used the proviral DNA was predominantly in CD32⁻ T cells.

10 HIV⁺ individuals were used to derive CD32⁺ and – CD4 T cells and qPCR used to measure proviral load, this recapitulates the experiment from Descours et al fig4b. The result from this analysis showed no statistical difference in proviral load between CD32⁻ and CD32⁺ and only 2/10 individuals had a higher load in CD32⁺ T cells. They conclude that the majority of infected CD4 T cells are CD32⁻.

This group were unable to perform viral out growth assays to be able to compare if this proviral population could give rise to reactivatable replication competent HIV.

Response: We have now evaluated replication competent HIV in the new Table 2.

This group conclude that their data suggest that CD32 expression identify activated CD4 T cells, but is not a marker of the latent reservoir. They discuss that their results show in 6/10 HIV⁺ donors tested that proviral copies/ cell were higher in CD32⁻ T cells and that this was comparable to Descours et al raw data in supplemental data (5/9 HIV⁺ donors), however it is unclear what specific reinterpretation of raw data that is being performed here? This requires clarification.

Response: We referred to the raw data presented as supplemental material in the Descours original article. This is now clearly stated in the revised manuscript.

The discussion is not clear, as to why the results from this publication are different from Descours et al., this clearly needs to be resolved.

Response: We clearly show that CD32 is a marker of activated cells, which may not be considered an HIV reservoir. Our data indicate also that CD32 positive cells do not significantly differ from CD32 negative cells in the number of viral DNA copies per cell and in the presence of replication competent viruses, being both evidences different from that of Descours et al. We have clarified the discussion in order to better reflect our findings. However, although we feel that there are a number of significant shortcomings in the work of Descours et al., it is not our aim to disqualify their work. We expect that further research by others and ourselves will provide adequate explanations to the role of CD32 in HIV-1 infection.

The ability of the latent reservoir to reactivate and produce replication competent virus is clearly an important attribute over and above just proviral load. Performing these experiments would be highly desirable in this situation and these experiments

should be performed, it needs to be addressed if CD32+ T cells contribute the majority of replication competent HIV or if again the CD32- T cell population is equally important in contributing to the reactivatable replication competent reservoir?

Response: We now show the evaluation of replication competent HIV in the new Table 2.

Minor points the English in the paper could benefit from editing to help with the flow and readability.

Response: The manuscript has been revised by the Springer Nature Editing service.

Reviewers' comments:

Reviewer #1 (Remarks to the Author):

I appreciate the author's answers to my comments. However, there are still several aspects in this manuscript that would need to be modified or clarified. The experiments presented in this manuscript clearly show a link between CD32 expression on CD4+ T cells and T cell activation. However, the virological data are much less convincing and the role of HIV infection in that process needs to be clarified. The new IUPM data are difficult to interpret and the IUPM values reported here are 3 log higher than in published studies. In addition, the integrated HIV DNA values are also surprisingly high (1 to 10% of infection in CD4+ T cells). The authors should consider the following remarks:

1. I still disagree on the first point, which is in my view, critical. Stating that the HIV reservoir identified by Descours et al. is "latent" is an incorrect interpretation of the original study by the authors (and apparently by other scientists who commented on this article in "news and views" as mentioned in the author's response.) In Descours et al., it was never said that the CD32+ T cells isolated from people on ART were latently infected. Accordingly, I disagree with the following sentence from the authors: "the HIV reservoir definition, i.e. cells (CD4 T cells or others) in which a replication-competent form of HIV persists, entails the existence of viral latency". Latency and HIV persistence are 2 different concepts. For instance, residual viral replication can sustain a persistent viral reservoir without requiring latency. Therefore, I think the manuscript should be modified to clearly distinguish these concepts. In the last sentence of the abstract ("These results raise questions regarding the immune resting status of CD32+ cells harboring HIV-1 proviruses"), the authors infer that Descours et al. investigated the "immune status" of the CD32+ cells which is not correct. The only data that directly contradicts the Descours findings are those showing no enrichment in HIV DNA (or replication competent HIV) in CD32+ cells, which is independent from the activation status of these cells. The first sentence of the abstract should also be corrected and all the manuscript should be modified to acknowledge that Descours et al. did not identify CD32+ CD4+ T cells as a "latent" reservoir. Indeed, the recent work from Abdel-Mohsen et al (Science Translational Medicine, 2018) confirms that CD32+ identifies a transcriptionally active reservoir and not a latent reservoir for HIV, further reinforcing the importance of distinguishing these 2 concepts.
2. Line 109: "HIV-1 infection induced CD32 expression in PHA/IL-2 activated CD4+ T cells (Fig. 3a)." When looking at the third dot plot, it looks like the majority of the CD32+ cells are found within the GFP negative population. Although efavirenz somewhat reduces CD32 expression, it is hard to distinguish the relative effect of HIV sensing and HIV infection in these experiments.
3. Line 128: "This finding indicates that CD32 expression is a marker of T cell activation." This conclusion applies to Figure 1, not to Figure 4.
4. The frequencies of infection measured by integrated HIV DNA (Figure 5b) are surprisingly high (1 to 10%). This is at least a log higher than expected.
5. The new experiment aimed at measuring replication competent HIV in CD32- and CD32+ populations is hard to interpret. The authors report a mean IUPM value of 39886 in Table 1. IUPM are usually in the range of 0.1 to 10. I don't think the numbers in Table 1 represent IUPM. Also, the 95%CI range do not overlap with the IUPM values. This should be clarified.
6. Finally, VOA sensitivity depends on the number of cells used to perform the assay. These should be indicated (particularly for the CD32+ fraction). It is surprising that the authors found positive culture in almost all CD32+ sorted populations given their very low frequency, and according to the authors, their similar frequency of infection compared to total CD4+ T cells. The number of cells in each well (500-20,000 cells) seems extremely low as well to measure replication competent HIV.

Reviewer #2 (Remarks to the Author):

Badia and colleagues have adequately addressed the issues raised in review. We appreciate the effort performing viral replication assays in populations of CD32+ vs CD32- cells. These assays demonstrate that the global level of infection in these cell populations is relatively similar. The authors should note that these assays are distinctly different from outgrowth assays of latent replication-competent HIV performed in resting cells as described by Siliciano and Siliciano.

Reviewer #3 (Remarks to the Author):

The authors have addressed all of my comments including requested additional experiments and I am now satisfied with the paper

Response to reviewers' comments

Reviewer #1 (Remarks to the Author):

The experiments presented in this manuscript clearly show a link between CD32 expression on CD4+ T cells and T cell activation.

Response: We thank the reviewer for this comment.

The new IUPM data are difficult to interpret and the IUPM values reported here are 3 log higher than in published studies. In addition, the integrated HIV DNA values are also surprisingly high (1 to 10% of infection in CD4+ T cells).

Response: As mentioned in Methods, we used a protocol that takes advantage of a reporter cell line (TZM-bl cells) to identify replication competent HIV. The use of this cell line as reporter has been shown to have a 1000-fold increase in sensitivity and helped to demonstrate that the size of the inducible latent HIV-1 reservoir in aviremic participants on therapy may be approximately 70-fold larger than previous estimates (Sanyal et al Nature Medicine, 2017).

Importantly, we compared equal number of CD32- and CD32+ cells, unlike Descours et al that compared CD32+ cells to total cells and their cell cultures differed in cell numbers per well. In our assay both CD32- and CD32+ cultures started from the same number of cells and followed identical procedure, so irrespective of the estimated size of the reservoir, relative results would indicate differences between CD32- and CD32+.

The authors should consider the following remarks:

I still disagree on the first point, which is in my view, critical. Stating that the HIV reservoir identified by Decours et al. is "latent" is an incorrect interpretation of the original study by the authors (and apparently by other scientists who commented on this article in "news and views" as mentioned in the author's response.)

In Descours et al., it was never said that the CD32+ T cells isolated from people on ART were latently infected. Accordingly, I disagree with the following sentence from the authors: "the HIV reservoir definition, i.e. cells (CD4 T cells or others) in which a replication-competent form of HIV persists, entails the existence of viral latency". Latency and HIV persistence are 2 different concepts. For instance, residual viral replication can sustain a persistent viral reservoir without requiring latency. Therefore, I think the manuscript should be modified to clearly distinguish these concepts.

Response: Following recommendations from the first revision, we refer to the work by Descours et al as "CD32 is recently proposed to be a marker of the CD4 T cell HIV reservoir" .We have now rechecked our manuscript to clearly avoid saying that the HIV reservoir identified in patients by Decours et al. is "latent" except when shown in vitro. However, in the Introduction, we use a well accepted definition of the HIV reservoir and

its corresponding reference. To further clarify the definition of HIV reservoir, we have now included additional text, taken from two additional references including one from the IAS Scientific Working Group on HIV Cure coauthored by Dr. M. Benkirane, senior author of the Descours paper.

In the last sentence of the abstract (“These results raise questions regarding the immune resting status of CD32+ cells harboring HIV-1 proviruses”), the authors infer that Descours et al. investigated the “immune status” of the CD32+ cells which is not correct.

Response: The sentence in the abstract refers to our results, not to the results of Descours et al. For clarity we have changed the text to “Our results....”

The only data that directly contradicts the Descours findings are those showing no enrichment in HIV DNA (or replication competent HIV) in CD32+ cells, which is independent from the activation status of these cells. The first sentence of the abstract should also be corrected and all the manuscript should be modified to acknowledge that Descours et al. did not identify CD32+ CD4+ T cells as a “latent” reservoir.

Response: We do not aim at contradicting Descours findings but to shed light on the role of CD32 expression in HIV infection. In this sense, and unlike Descours et al., we demonstrate that CD32 does not mark for an HIV latent reservoir in HIV+ individuals, which is a significant advancement. Thus, the current version of our manuscript only refers to “latent” reservoir when discussing our own data. However, to avoid misunderstandings we have modified the first sentence of the abstract to literally cite Descours et al: “CD32a has been shown to be preferentially expressed in latently HIV-1 cells, using an in vitro model of infected quiescent CD4 T cells”.

Indeed, the recent work from Abdel-Mohsen et al (Science Translational Medicine, 2018) confirms that CD32+ identifies a transcriptionally active reservoir and not a latent reservoir for HIV, further reinforcing the importance of distinguishing these 2 concepts.

Response: We have now included the reference by Abdel-Mohsen et al (Science Translational Medicine, 2018). We literally cite their own words: Abdel-Mohsen et al clearly state: 1. “These results challenge the notion that CD32 enriches for HIV latently infected cells” 2. “Immunoprofiling of CD32+ CD4+ T cells in blood and tissues of humans and RMs shows that these cells exhibit an activated and differentiated phenotype, making it unlikely that they are enriched with HIV latently infected cells”.

2. Line 109: “HIV-1 infection induced CD32 expression in PHA/IL-2 activated CD4+ T cells (Fig. 3a).” When looking at the third dot plot, it looks like the majority of the CD32+ cells are found within the GFP negative population. Although efavirenz somewhat reduces

CD32 expression, it is hard to distinguish the relative effect of HIV sensing and HIV infection in these experiments.

Response: As seen in the bar graph of Figure 3a, the ratio of infected cells is not different between CD32+ and CD32- populations. In addition, the results with the HIV inhibitor efavirenz clearly demonstrate that changes in CD32 expression are dependent on virus replication. These results are in line with Descours et al. that indicated that CD32 expression was dependent on virus replication and blocked by the HIV inhibitor raltegravir. We did not aim at measuring HIV sensing and thus we do not explore this issue.

3. Line 128: "This finding indicates that CD32 expression is a marker of T cell activation." This conclusion applies to Figure 1, not to Figure 4.

Response: Fig. 4e shows the expression of activation markers HLA-DR+/CD69+ cells for each of the 5 donors used in Fig. 4d. The sentence has been modified so it is clearly associated to data shown in Fig. 4.

4. The frequencies of infection measured by integrated HIV DNA (Figure 5b) are surprisingly high (1 to 10%). This is at least a log higher than expected.

Response: Fig. 5b shows as few as one integrated provirus in one thousand cells (0,001 copies/cell) and up to two integrated provirus/cell. This is well in line with other reports, including that of Descours et al. that shows a number of patients with HIV DNA/cells at roughly 0,001 copies/cell and up to 3 HIV DNA copies/cell in one patient (Fig. 3b in their publication).

5. The new experiment aimed at measuring replication competent HIV in CD32- and CD32+ populations is hard to interpret. The authors report a mean IUPM value of 39886 in Table 1. IUPM are usually in the range of 0.1 to 10. I don't think the numbers in Table 1 represent IUPM. Also, the 95%CI range do not overlap with the IUPM values. This should be clarified.

Response: Our interpretation is unambiguous: there are no significant differences in the mean values between CD32- and CD32+ cells. The values represent the maximum likelihood estimate of infection frequency (in infectious units per million) as indicated in the IUPMStats v1.0 Infection Frequency Calculator (<http://silicianolab.johnshopkins.edu>). This is now clearly mentioned in Methods.

As state above, differences in IUPM values may be due to the use of a more sensitive TZM reporter cell line and indeed, Descours et al, report IUPM values ranging from 2,2 to 16422 being not that different to our estimates.

Table 2 with 95% CI ranges was corrected for a typing error.

6. Finally, VOA sensitivity depends on the number of cells used to perform the assay. These should be indicated (particularly for the CD32+ fraction). It is surprising that the authors found positive culture in almost all CD32+ sorted populations given their very low frequency, and according to the authors, their similar frequency of infection compared to total CD4+ T cells. The number of cells in each well (500-20,000 cells) seems extremely low as well to measure replication competent HIV.

Response: We did not compare CD32+ cells to total CD4+ T cells. We did a head to head comparison of equal number of CD32- and CD32+ cells (mentioned in Methods). So differences, if any, could also be compared relatively to each other (CD32- vs. CD32+). As mentioned in the text, we used an adapted, more sensitive TZM reporter cell line for virus titrations, following a 21 day coculture. Not all cultures were found positive. In fact, Table 2 shows a subset of 10 individuals out of an expanded cohort of 23.

Reviewer #2 (Remarks to the Author):

Badia and colleagues have adequately addressed the issues raised in review. We appreciate the effort performing viral replication assays in populations of CD32+ vs CD32- cells. These assays demonstrate that the global level of infection in these cell populations is relatively similar. The authors should note that these assays are distinctly different from outgrowth assays of latent replication-competent HIV performed in resting cells as described by Siliciano and Siliciano.

Response: We thank the reviewer for the impartial review of our manuscript. References from which our assay was adapted are included in Methods.

Reviewer #3 (Remarks to the Author):

The authors have addressed all of my comments including requested additional experiments and I am now satisfied with the paper

Response: We thank the reviewer for the impartial review of our manuscript.

REVIEWERS' COMMENTS:

Reviewer #1 (Remarks to the Author):

I am not satisfied by the author's response for the following reasons:

1. A median IUPM value of 40,000 as reported in Table 2 contradicts hundreds of publications that have measured replication competent HIV in patient's samples. The authors claim that they used the method of Sanya et al: While it is true that this assay may be more sensitive than the classical QVOA, Sanya et al reported an average IUPM value of 46.9, which is almost 3 logs lower than what the authors indicate in Table 2.

2. I am still not convinced by the HIV DNA data and the explanation provided by the authors is not reassuring. In the CD32negative fraction, Descours et al reported DNA value ranging from 0.0001 to 0.01 copies/cell whereas the authors report values between 0.01 to 1 copies/cell. There is a 2 log difference between these measures. If the authors were correct, 1 to 10% of all CD4+ T cells would contain HIV DNA (assuming that the majority of infected cells harbor a single genome, as previously demonstrated), which is 100 to 1,000 fold higher than in any study in which the reservoir has been measured by PCR.

Reviewer #2 (Remarks to the Author):

In this revision, the authors have attempted to answer the concerns raised by a reviewer. These discussion points focus on semantics of what was written and demonstrated in the Descours et al work. I agree with the authors that Descours et al stated ".....surface expression in HIV-infected quiescent CD4 T cells shows...CD32a, is the most highly induced, with no detectable expression in bystander cells....Using blood samples from HIV-1-positive participants receiving suppressive antiretroviral therapy, we identify a subpopulation of 0.012% of CD4 T cells that express CD32a and host up to three copies of HIV DNA per cell. This CD32a+ reservoir was highly enriched in inducible replication-competent proviruses...Our discovery that CD32a+ lymphocytes represent the elusive HIV-1 reservoir may lead to insights....."

However the authors of this paper clearly show shortcomings of the claims and findings of this initial work, most clearly in Table 2, wherein there is no enrichment seen in the CD32a+ vs CD32a- populations.

Again I would suggest that the authors clarify the findings of Table 2, as this assay registers the entry into a cell of particles that must only produce Tat to register positive in the TZMBL assay, but may be still defective in numerous ways that prevents serial passage of replication-competent HIV. This is most likely to be the explanation for the fact that the IUPM frequency is much higher than that recorded by many groups in quiescent, persistently infected CD4+ T cells. Nevertheless, these results show a lack of enrichment in viral recovery in the CD32a+ population.

** See Nature Research's author and referees' website at www.nature.com/authors for information about policies, services and author benefits

Response to reviewers' comments

Reviewer #1

1. A median IUPM value of 40,000 as reported in Table 2 contradicts hundreds of publications that have measured replication competent HIV in patient's samples. The authors claim that they used the method of Sanya et al: While it is true that this assay may be more sensitive than the classical QVOA, Sanya et al reported an average IUPM value of 46.9, which is almost 3 logs lower than what the authors indicate in Table 2.

Response: Kindly, see the response to Reviewer #2.

2. I am still not convinced by the HIV DNA data and the explanation provided by the authors is not reassuring. In the CD32negative fraction, Descours et al reported DNA value ranging from 0.0001 to 0.01 copies/cell whereas the authors report values between 0.01 to 1 copies/cell. There is a 2 log difference between these measures. If the authors were correct, 1 to 10% of all CD4+ T cells would contain HIV DNA (assuming that the majority of infected cells harbor a single genome, as previously demonstrated), which is 100 to 1,000 fold higher than in any study in which the reservoir has been measured by PCR

Response: In our manuscript we will not enter into further disqualifying the paper by Descours et al. The range of DNA determinations in our assay fall within the range of Descours et al. and the reviewer focuses on a particular cell subset. The limited number of samples evaluated by Descours et al. or in our study may account for the differences. However, like Adbel-Mohsen et al. we do not find significant differences between CD32+ and CD32- cells.

Reviewer #2 (Remarks to the Author):

Again I would suggest that the authors clarify the findings of Table 2, as this assay registers the entry in to a cell of particles that must only produce Tat to register positive in the TZMBL assay, but may be still defective in numerous ways that prevents serial passage of replication-competent HIV. This is most likely to be the explanation for the fact that the IUPM frequency is much higher than that recorded by many groups in quiescent, persistently infected CD4+ T cells. Nevertheless, these results show a lack of enrichment in viral recovery in the CD32a+ population.

Response: We agree with the reviewer. We have added text to clarify this possibility, that is, overestimation of replication-competent virus due to HIV-1 Tat expression in Tzm-bl cells:

Page 7, Results: Co-culture supernatants were titrated in CD4+ Tzm-bl cells to evaluate the replication competence of the amplified virus, which was measured as luciferase production. In this model, released virus from CD32+ or CD32- CD4+ T cells should be competent enough to enter target cells and at least mediate Tat-dependent luciferase expression.

Page 11, Discussion: "The TZM-bl assay used in our study records virus that must only produce Tat upon entry into cells to register a positive signal but may still be defective in numerous ways. Thus, the assay may be overestimating replication-competent HIV, explaining the higher IUPM frequency observed in our study than that recorded by many groups in quiescent, persistently infected CD4+ T cells. However, we compared the viral outgrowth of cultures with an equal cell number for CD32- and CD32+ cells, allowing for head-to-head comparisons between both cell types"